# Chitin-Based Magnesium Oxide Biocomposite for the Removal of Methyl Orange from Water

**DOI:** 10.3390/ijerph20010831

**Published:** 2023-01-01

**Authors:** Hicham Majdoubi, Ayoub Abdullah Alqadami, Rachid EL Kaim Billah, Marta Otero, Byong-Hun Jeon, Hassan Hannache, Youssef Tamraoui, Moonis Ali Khan

**Affiliations:** 1Materials Science Energy and Nanoengineering Department (MSN), Mohammed VI Polytechnic University (UM6P), Lot 660-Hay Moulay Rachid, Benguerir 43150, Morocco; 2Chemistry Department, College of Science, King Saud University, Riyadh 11451, Saudi Arabia; 3Laboratory of Coordination and Analytical Chemistry, Department of Chemistry, Faculty of Sciences, University of Chouaib Doukkali, Avenue Jabran Khalil Jabran, B.P 299, El Jadida 24000, Morocco; 4Departmento de Química y Física Aplicadas, Universidad de León, Campus de Vegazana s/n, 24071 León, Spain; 5Department of Earth Resources & Environmental Engineering, Hanyang University, 222 Wangsimni-ro, Seongdong-gu, Seoul 04763, Republic of Korea; 6Laboratory of Engineering and Materials LIMAT, Faculty of Science Ben M’Sik, Hassan II University, Casablanca 2600, Morocco

**Keywords:** sea food waste, waste management, anionic dye, adsorption, regeneration

## Abstract

In this work, a cost-effective chitin-based magnesium oxide (CHt@MgO) biocomposite with excellent anionic methyl orange (MO) dye removal efficiency from water was developed. The CHt@MgO biocomposite was characterized by FT-IR, XRD, SEM-EDX, and TGA/DTG. Results proved the successful synthesis of CHt@MgO biocomposite. Adsorption of MO on the CHt@MgO biocomposite was optimized by varying experimental conditions such as pH, amount of adsorbent (m), contact time (t), temperature (T), and initial MO concentration (*C_o_*). The optimized parameters for MO removal by CHt@MgO biocomposite were as follows: pH, 6; m, 2 g/L; t, 120 min. Two common isotherm models (Langmuir and Freundlich) and three kinetic models (pseudo-first-order (PFO), pseudo-second-order (PSO), and intraparticle diffusion (IPD)) were tested for experimental data fitting. Results showed that Langmuir and PFO were the most suitable to respectively describe equilibrium and kinetic results on the adsorption of MO adsorption on CHt@MgO biocomposite. The maximum Langmuir monolayer adsorption capacity (*q_m_*) on CHt@MgO biocomposite toward MO dye was 252 mg/g at 60 °C. The reusability tests revealed that CHt@MgO biocomposite possessed high (90.7%) removal efficiency after the fifth regeneration cycle.

## 1. Introduction

Effluents from the textile, pharmaceutical, and paper and pulp industries are the major carriers of dyes to the environment. These dyes create an overall imbalance in the aquatic ecosystem by inhibiting sunlight penetration [1]. Annually, more than 7000 tons of synthetic dyes are being manufactured [2], and it has been estimated that ~100 tons of untreated color-containing effluents are discharged into water bodies [3]. Methyl orange (MO, C_14_H_14_N_3_NaO_3_S) is an anionic dye with an azo (−N=N−) group. It is widely used as a coloring agent for several industrial applications, an acid–base indicator for titrations, and a biological dye [2]. The presence of the −N=N− group and benzene rings in MO molecule and its low biodegradability make its presence in aquatic systems an issue of concern for the environmental and human health [3]. Therefore, it is necessary to remove MO from wastewater before its discharge to water reservoirs.

Photocatalytic degradation [4], advanced oxidation technology [5], membrane separation [6], reverse osmosis [7], and adsorption [8] are among the most commonly used processes for the removal of anionic dyes from wastewater. Among them, adsorption is advantageous because it is a highly efficient cost-effective process, with operational ease and the possibility of after-use adsorbent regeneration and reutilization. Different novel adsorbents such as activated carbon polyaniline@BiVO_4_ [9], MOF-235 [10], carbon-coated monolith [11], and acid-modified carbon-coated monolith [12] have been successfully used for the removal of MO dye. However, as stated above, the application feasibility of an adsorption process primarily depends on the operational costs and the removal performance of adsorbents used. Therefore, researchers are looking for adsorbents that are economically feasible, ecological safe, insensitive to toxic substances, and have high removal efficiency.

Chitin (CHt, poly(β-(1–4)-*N*-acetyl-D-glucosamine)), which is the second most abundant biopolymer on Earth (after cellulose), is mainly found in crustacean shells, insects, microorganisms such as algae, and yeasts [13]. Shrimp shell is composed of Cht (20–30%), minerals (30–50%), and protein (30–40%) [14]. At global scale, 6–8 million tons/annum of crab, shrimp, and lobster shell waste is generated from the seafood processing industries [15], which creates a major disposal issue in coastal areas. CHt is a nontoxic, biodegradable, biocompatible biomaterial with intrinsic rich N- and O-containing functional groups [16]. These properties make it a highly efficient adsorbent for water remediation applications. Acid Blue 25 dye was removed from aqueous solution using *Penaeus indicus* shrimp [17]. Direct Red 80 and Direct Blue 71 dyes were removed from water using nontreated shrimp shells [18]. He et al. [19] synthesized waste shrimp shell hydrochar for the removal of MO dye from an aqueous solution with a maximum adsorption capacity of 755 mg/g at an optimal pH of 4.0.

CHt is insoluble in most solvents owing to its dense structure; therefore, its chemical modification is feasible. Combining CHt with different biopolymers, organics and inorganics, and carbon-based materials may enhance its adsorption performance toward dyes. Therefore, different CHt-based adsorbents have been developed for the removal of dyes from contaminated waters such as CHt hydrogel (CG3) [20], CHt beads [21], CHt/lignin [22], magnetic graphene oxide/CHt [23], chitin-cl-poly(itaconic acid-co-acrylamide)/zirconium tungstate nanocomposite [24], nitrogen-enriched carbon nanofiber aerogels [25], and CHt/CS-g-PAM [26].

Magnesium oxide (MgO) is commonly used as an additive in refractory products and paints, as well as in toxic waste remediation, in catalysis, and as a bactericide [27,28]. Due to its chemical stability and nontoxicity, it is a promising material in water purification [29]. Hu et al. [30] developed MgO nanoplates for Congo red and reactive brilliant red X3B dye adsorption, with maximum adsorption capacities of 303.0 and 277.8 mg/g, respectively. In two separate studies on the use of nanostructured MgO as adsorbent, the observed adsorption capacity of reactive blue 19 dye was 250 mg/g [31], while 166.7 and 123.5 mg/g were the reported capacities for the adsorption of reactive blue 19 and reactive red198, respectively [32]. Therefore, it is expected that composting of CHt with MgO might enhance its anionic dye adsorption potential.

In the described context, the main objectives of this study were to extract CHt from waste shrimp shells through the two-step process of demineralization and deproteinization, and then composting it with MgO to develop CHt@MgO with high adsorption capacity for the removal of MO dye from aqueous solutions. The CHt@MgO composite was characterized through FT-IR, XRD, SEM-EDX, and TGA/DTG analyses. The effects of experimental parameters such as adsorbent mass, solution pH, contact time, temperature, and initial MO concentration on the adsorption process were investigated through batch mode experiments. Additionally, the adsorption kinetics, isotherms, and thermodynamics of the adsorption of MO onto CHt@MgO were assessed.

## 2. Experimental

### 2.1. Chemicals and Reagents

Shrimp shells were collected from a local market in Casablanca (Morocco). Sodium hydroxide (NaOH, Sigma-Aldrich, St. Louis, MO, USA, 99.8%), hydrochloric acid (HCl, Sigma-Aldrich, 37%), 1,4-dioxane extra pure (Sigma-Aldrich, 99.8%), glacial acetic acid (Sigma-Aldrich, 99.7%), magnesium oxide (MgO, Sigma-Aldrich, 99.99%), and methyl orange (MO, Sigma-Aldrich, 99%) (MO structure is shown in Figure 1) were used during the study.

### 2.2. Preparation of CHt@MgO

#### 2.2.1. Extraction of Chitin

Chitin from shrimp shells was extracted following a previously reported two-step method involving demineralization and deproteinization steps [1,13] (Figure 2). Shrimp shells were washed and dried. Twenty of the dried shrimp shells were crushed to powder. Then, 20 g of shrimp shell powder was demineralized at room temperature using 1 N HCl solution (*w/v* 1:20). The mixture was mechanically stirred for 6 h at 25 °C. The obtained product was washed with deionized (D.I) water to neutral pH and then dried for 24 h at 60 °C. Thereafter, deproteinization of demineralized powder was carried out through refluxing in 5% NaOH solution (*w/v* 1:20) at 80 °C under continuous stirring for 10 h. The CHt was separated by filtration, washed with DI water to neutral pH, and dried at 60 °C for 24 h.

#### 2.2.2. Synthesis of CHt@MgO

The extracted CHt powder (3.5 g) was ultrasonically dispersed in 30 mL of 1,4-dioxane solution. Simultaneously, 1 g of MgO was ultrasonically dispersed in 20 mL of 1,4-dioxane solution. A composite was prepared by mixing CHt and MgO dispersions under continuous stirring at 60 °C for 30 min (Figure 2). The mixture was then casted on siliconized paper and dried in an oven at 50 °C for 3 h to obtain CHt@MgO.

### 2.3. Characterization of CHt@MgO

X-ray diffraction (XRD) analysis was carried out using a D8 diffractometer from Bruker (Billerica, Massachusetts, USA) operating at 45 kV/100 mA, using CuKα radiation with a Ni filter. The surface morphology of the samples was analyzed using a scanning electron microscope (SEM) Philips XL 30 ESEM (Acc spot Magn 20.00 kV). Fourier-transform infrared (FT-IR) spectroscopy (Thermo Fisher Scientific (Waltham, MA, USA) Spectrometer) was used to detect active functional groups over composite surfaces. Thermogravimetric analysis (TGA) was performed using a Discovery TGA from TA instruments (Waters Corporation, Milford, MA, USA) at a heating rate of 10 °C/min under N_2_ atmosphere.

### 2.4. MO Adsorption and Regeneration Studies

Adsorption experiments of MO on CHt@MgO biocomposites were performed in batch mode. Erlenmeyer flasks (150 mL), each containing 100 mL of MO solution with an initial concentration 100 mg/L at pH 6 was equilibrated with 0.2 g of CHt@MgO at 300 rpm stirring speed. The adsorption kinetics of MO was studied at varied MO initial concentrations (100, 200, and 300 mg/L). Adsorption isotherms were studied by varying MO initial concentrations between 25 and 400 mg/L at varied temperatures (25, 40, and 60 °C). Furthermore, the effect of temperature on MO adsorption was investigated at 25, 40, and 60 °C under the following experimental conditions: pH 6, 50 mL of MO aqueous solution with an initial MO concentration of 100 mg/L, and 2 g/L of CHt@MgO. The adsorbed concentration of MO at equilibrium (*q_e_*, mg/g) and at time t (*q_t_*, mg/g), and the percentage adsorption were estimated using the following equations:(1)$$q_{e}=\left(C_{o}-C_{e}\right)\times \frac{V}{m},$$
(2)$$q_{t}=\left(C_{o}-C_{t}\right)\times \frac{V}{m},$$
(3)$$\%adsorption=\frac{C_{o}-C_{e}}{C_{o}}\times 100,$$
where *m* (g) is the mass of CHt@MgO, *V* (L) is the volume of solution, and *C_o_* (mg/L), *C_e_* (mg/L), and *C_t_* (mg/L) represent the initial, equilibrium, and time *t* concentrations of MO dye in aqueous solution, respectively.

The regeneration of CHt@MgO and its subsequent reutilization for MO adsorption were also studied in batch mode. CHt@MgO (0.1 g) was saturated for 1 h with 50 mL of 100 mg/L MO solution. Thereafter, the MO-saturated CHt@MgO was separated, and then rinsed several times with deionized (D.I) water to remove unadsorbed MO traces. The MO-saturated CHt@MgO was treated with 0.5 M NaOH solution to desorb MO. The process was repeated for seven consecutive regeneration cycles.

## 3. Results and Discussion

### 3.1. Characterization of CHt@MgO

Functional groups present on CHt, CHt@MgO, and MgO samples were determined by FT-IR, and the results are illustrated in Figure 1a. In the FT-IR spectrum of CHt, the broad peaks between 3256 and 3437 cm^−1^ were due to O–H stretching and N–H groups on the surface. The peaks at 2962, 1649, 1590, 1414, and 1374 cm^−1^ were attributed to C–H sp^3^ vibration, stretching vibrations of C=O (amide I), bending vibrations of amide II (N–H) [19], twisting vibrational stretch of CH_2_, and C–N stretching, respectively [33]. The peaks in the range from 1154 to 1032 cm^−1^ were due to the stretching vibration of C–O–C and C-O bonds in the structure [34]. A peak of –CH_3_ in acrylamide groups was observed at 885 cm^−1^ [35]. The characteristic peak of β-1,4 glycosidic, and OH out-of-plane binding were present at 933 and 670 cm^−1^, respectively [19]. All these peaks confirmed successful extraction of CHt from shrimp shells. In the FT-IR spectrum of MgO, the major peaks at around 594, 557, 565, and 544 cm^−1^ indicated the Mg–O vibrations of MgO [36]. Peaks related to the –OH groups of adsorbed water from the atmosphere on the surface of MgO were observed at 3699 cm^−1^ and 1414 cm^−1^ [37]. A peak at 855 cm^−1^ was characteristic of cubic MgO [37]. The FT-IR spectrum of CHt@MgO composite (Figure 1a) exhibited peaks at 3698 and 3310 cm^−1^ attributed to the stretching vibration mode of –OH and –NH groups, respectively. The peaks of C–H symmetric and asymmetric stretching vibration appeared at 2928 and 2855 cm^−1^, respectively. The peaks at 1644, 1561, 1414, and 1025–1059 cm^−1^ were due to the C=O amide stretching vibration, –NH amide bending vibration, C–N axial deformation, and C–O–C stretching vibration, respectively [19]. The characteristic peaks for MgO appeared at higher wave numbers (650, 601, and 564 cm^−1^) compared to those in the FT-IR spectrum of pure MgO [37]. The spectral results confirmed the successful production of the CHt@MgO biocomposite.

The X-ray diffraction (XRD) patterns of CHt, CHt@MgO, and MgO samples are presented in Figure 1b. In the XRD pattern of CHt, three characteristic diffraction peaks at 2θ = 9.6°, 19.30°, and 21,95°, respectively corresponding to (0 2 0), (1 1 0), and (1 2 0) crystal planes, were observed, which revealed the crystalline and amorphous structure [33] of CHt, in agreement with previously reported results [38]. The diffraction peaks of CHt were present in the XRD pattern of CHt@MgO, along with new diffraction peaks at 2θ = 37.76° (1 1 1), 42.73° (2 0 0), 58.45° (1 1 0), 50.5° (1 0 2), and 62.13° (2 2 0), which matched the cubic lattice of MgO reported in the literature [28]. The pure MgO diffraction peaks appeared at 2θ = 18.50°, 37.76°, 42.73°, 58.45°, 50.5°, and 62.13°, as attributed to (0 0 1), (1 1 1), (2 0 0), (1 1 0), (1 0 2), and (2 2 0) planes of MgO [37]. Compositing with CHt shielded the MgO peak intensities. Thus, low-intensity MgO peaks were observed in CHt@MgO patterns.

The thermogravimetric results (DTG/TGA) obtained for CHt and CHt@MgO composite are displayed in Figure 1 c and d, respectively. As can be observed in Figure 1c, the major decomposition peak of CHt@MgO occurred at a slightly lower temperature than that of CHt, but the percentage mass loss of the composite was lower than that of CHt, which may be related to the relatively high thermal stability of MgO [39]. Therefore, the total mass loss was 60% and 54% for CHt and the CHt@MgO composite, respectively (Figure 1d). The thermal decomposition profile of CHt exhibited three stages of mass loss (Figure 1c,d). During the first stage, ~6% mass loss arose in the temperature range 30–100 °C owing to the loss of surface-absorbed moisture. The second stage of mass loss (~7%), in the range of 190–400 °C and peaking at 250 °C, was due to the decomposition of functional groups of CHt. The third stage of mass loss (~47%) occurred in the temperature range of 430–700 °C, which may be attributed to the degradation of the saccharide ring of CHt [40,41]. The DTG and TGA curves of CHt@MgO biocomposite also showed thermal decomposition in three stages (Figure 1c,d). The first stage of mass loss (~2%) occurred at <100 °C, which may be assigned to the loss of water [42]. The second and third stages involved mass losses of 8% and 44% in temperature ranges of 170–400 °C (peaked at 230 °C) and 400–700 °C, respectively, which were due to the decomposition of functional groups and degradation of the saccharide ring of CHt, respectively [41].

The morphology of CHt showed an irregular surface with different grain sizes and internal pore structures (Figure 2a). After modifying CHt with MgO, due to the dispersion of MgO particles onto the CHt surface, the surface of the resultant CHt@MgO became rough and showed apparent pores and cavities (Figure 2b). The results of the elemental analysis through EDX showed that CHt contains carbon (65.62%), oxygen (32.40%), and nitrogen (1.98%) (Figure 2c). The weight percentage of these elements changed after the modification of CHt with MgO. Furthermore, new characteristic peaks of Mg (35.5%) appeared, indicating that MgO became well bonded with CHt during compositing (Figure 2d).

### 3.2. MO Adsorption Experiments on CHt@MgO Biocomposite

The pH plays an essential role in the surface charge of the MO molecule and of the CHt@MgO adsorbent, due to the associated changes in the attached functional groups. Therefore, the influence of pH on the adsorption of MO onto CHt@MgO was studied in the pH range 2–12, and the obtained results are illustrated in Figure 3a. Under the experimental conditions used, it was observed that the MO adsorbed concentration in the equilibrium (*q_e_*, mg/g) onto CHt@MgO increased from 44.7 to 48.35 mg/g as the pH increased from 2 to 3 due to electrostatic attraction interactions between the protonated adsorption sites of CHt@MgO and the anionic MO dye (pK_a_ = 3.46). From pH 3 to 7, *q_e_* remained almost constant, thereafter dropping until reaching a value of 22.2 mg/g at pH 12. Under acidic conditions, MO molecule is a zwitterion that carries both positive (^+^NH(CH_3_)_2_, ^+^N(CH_3_)_2_, or –N=N^+^H–) and negative (–SO_3_^−^), charges while CHt@MgO has a positive charge (^+^OH_2_). Therefore, electrostatic attraction interactions between –SO_3_^−^ on MO surface and ^+^OH_2_ on the CHt@MgO surface occurred, which favored the adsorption of CHt@MgO toward MO dye [2]. Evaluation of the CHt@MgO surface charge at different pH revealed that its zero-point charge (pH_pzc_) was ~6.7 (Figure 3a, inset). This indicates that the CHt@MgO surface was positively charged when pH <6.7 and negatively charged at pH > 6.7. Thus, at alkaline pH (above pH_zpc_), both the CHt@MgO biocomposite and MO were anionic, and electrostatic repulsion between negative charges of (–SO_3_^−^) on the MO surface and negative charges of (^−^OH) on the CHt@MgO surface caused the observed decrease in *q_e_* [43]. A similar trend was observed for the adsorption of MO dye on La/Co-SC adsorbent [44]. Thus, on the basis of the results in Figure 3a, pH 6 was chosen as the most favorable pH for subsequent MO adsorption experiments in the present study.

The influence of the CHt@MgO biocomposite amount (*m*) on the adsorption of MO (V: 50 mL, *C_o_*: 100 mg/L, pH: 6, T: 25 °C) was tested in range 0.5–5 g/L, and the obtained results are displayed in Figure 3b. As can be seen, an increase in the mass of CHt@MgO from 0.5 to 2 g/L resulted in an increase in MO% adsorption from 16% to 96%. Furthermore, the increase in CHt@MgO biocomposite amount between 2 and 5 g/L showed no change in the removal percentage, which remained at 96%. The increase from 16% to 96% in the adsorption of MO with the increase in CHt@MgO biocomposite mass was due to the provision of more reactive adsorption sites, which allowed for the adsorption of more MO dye molecules. Conversely, the *q_e_* (mg/g) went down as CHt@MgO biocomposite concentration increased. This phenomenon can be interpreted by the fact that, as may be seen in Equation (1), *q_e_* (mg/g) is inversely proportional to the adsorbent mass (*m*) [45]. Thus, 2 g/L of CHt@MgO was chosen as the adsorbent dosage for subsequent studies on MO adsorption.

The effect of contact time on the adsorption process was studied in the time range 5–180 min while keeping all other parameters constant (V: 50 mL, *C_o_*: 100 mg/L, *m*: 2 g/L, pH: 6, T: 25 °C); the results are displayed in Figure 3c. It can be observed that the adsorption capacity of CHt@MgO biocomposite toward MO dye rose dramatically during initial 30 min, and then gradually increased before attaining equilibrium between 100 to 150 min at varied MO concentrations. More than 82%, 63%, and 73% of the MO adsorption was reached within 30 min at initial MO concentrations of 100, 200, and 300 mg/L, respectively. The fast increase in *q_t_* (mg/g) at the initial times was attributed to the availability of a large number of vacant active adsorption sites on the CHt@MgO biocomposite surface. At equilibrium (i.e., after 120 min), the *q_e_* (mg/g) values for the adsorption of MO on the CHt@MgO biocomposite were 49.6, 88.6, and 105.7 mg/g at initial MO concentrations of 100, 200, and 300 mg/L. Therefore, 120 min was chosen as the equilibrium time for MO in subsequent tests.

The influence of initial MO concentration (*C_o_* between 25 and 400 mg/L) on adsorption was studied at 25, 40, and 60 °C, and the corresponding curves are displayed in Figure 3d. The results revealed that *q_e_* (mg/g) increased from 12.2 to 111.3 mg/g at 25 °C with the increase in *C_o_*, which could be related to *q_e_* (mg/g) having a directly proportional relationship with *C_o_* (Equation (1)). Furthermore, the increase in *C_o_* in the solution assisted in overcoming the mass transfer resistance between the solid phase (CHt@MgO) and aqueous phase (MO solution) with the provision of a driving force. In addition, results on the influence of varying temperatures ranging from 25 to 60 °C on MO dye adsorption onto CHt@MgO biocomposite are depicted in Figure 3d. It can be noticed that the MO uptake by CHt@MgO biocomposite rose with temperature, which was more evident with the increase in *C_o_*. For example, from 25 to 60 °C, *q_e_* (mg/g) increased from 12.25 to 12.40 mg/g at *C_o_* = 25 mg/L and from 113.3 to 124 mg/g at *C_o_* = 400 mg/L, indicating that the adsorption MO by CHt@MgO bioadsorbent is an endothermic process. The increase in adsorption with temperature from 25 to 60 °C may be due to an increase in the diffusion rate of MO on CHt@MgO surface and a decrease in the solution viscosity. An experimental maximum uptake of 124 mg/g MO onto CHt@MgO was observed at 60 °C at a *C_o_* = 400 mg/L. Similar observations were reported for the adsorption of MO and MB on MOF-235 [10], as well as for MO adsorption on NPCs-0.5-800 [46].

### 3.3. MO Adsorption Modelling

#### 3.3.1. Adsorption Isotherm Modeling

Equilibrium data were fitted to two common nonlinear isotherm models, namely, the Langmuir [47] (Equation (4)), and Freundlich [48] (Equation (5)) isotherm models.
(4)$$q_{e}=\frac{q_{m}K_{L}C_{e}}{1+K_{L}C_{e}},$$
(5)$$q_{e}=K_{F}\;C_{e}{}^{1/n},$$
where *q_m_* (mg/g) represents the maximum monolayer adsorption capacity of CHt@MgO, and *K_L_* (L/mg) and *K_F_* ((mg/g) (L/mg)^1/n^) are the Langmuir and Freundlich constants, respectively. The dimensionless exponent *n* is the adsorption intensity.

The nature of adsorption was determined by the separation factor (*R_L_*), a constant, expressed as:(6)$$R_{L}=\frac{1}{1+K_{L}C_{o}},$$
where *K_L_* represents the Langmuir constant (L/mg), and *C_o_* (mg/L) is the lowest initial concentration of MO dye. The adsorption process is unfavorable if *R_L_* > 1, favorable if 0 < *R_L_* < 1, linear if *R_L_* = 1, and irreversible if *R_L_* = 0.

The adsorption isotherm parameters and the fitting of experimental data are presented in Table 1 and Figure 4a. The correlation coefficient (R^2^) values and isotherm plots revealed that the data fitted the Langmuir isotherm model. This suggests a monolayer coverage of MO dye molecules over the CHt@MgO surface at equilibrium. The *q_m_* was found to be 251.62 mg/g at 60 °C. This value was comparable or greater than the *q_m_* values determined for the adsorption of MO by different adsorbents in the literature, as may be seen in Table 2 [1,2,49,50,51,52,53]. The values of *K_L_* increased from 0.003 to 0.022 L/mg with increasing temperatures from 25 to 60 °C, indicating that the adsorption intensity of MO onto CHt@MgO was enhanced with temperature. In addition, the *R_L_* values for MO dye adsorption onto CHt@MgO biocomposite were in the range 0.645–0.952, confirming that MO adsorption onto CHt@MgO is a favorable process [54].

#### 3.3.2. Kinetic Modeling

Kinetic models, namely, pseudo-first-order (PFO) (Equation (7)) [55], pseudo-second-order (PSO) (Equation (8)), and intraparticle diffusion (IPD) (Equation (9)) [56], were used to investigate the adsorption kinetics of MO dye onto CHt@MgO biocomposite.
(7)$$q_{t}=q_{e}\left(1-e^{-kt}\right),$$
(8)$$q_{t}=\frac{q_{e}^{2}k_{2}t}{1+q_{e\;}k_{2\;}t},$$
(9)$$q_{t}=k_{id\;}t^{0.5}+C,$$
where *k_1_* (1/min), *k_2_* (g/(mg·min)), and *k_id_* (mg/(g·min^1/2^)) are the PSO, PSO, and IPD rate constants, respectively, and *C* (mg/g) is the boundary layer thickness.

Fittings of the experimental kinetic results on the adsorption of MO onto CHt@MgO biocomposite are presented in Figure 4b, while the fitted parameters are depicted in Table 3. According to the R^2^ values and kinetic plots, PFO was, among the considered models, the most adequate to describe MO adsorption data on CHt@MgO. This was confirmed by the closeness between the *q_e,exp_* and the q_e,cal._ The description of kinetic results by the PFO model suggests that the MO binding interaction with CHt@MgO biocomposite surface was physical adsorption involving electrostatic interaction between MO molecules and the CHt@MgO bioadsorbent. According to the IPD model (Table 3), the fitted intercept (*C*) value at varied concentrations ranged between 13.23 and 17.29 mg/g, which indicate that the IPD was not the only rate-controlling step [57].

#### 3.3.3. Thermodynamic Modeling

The Gibbs free energy change (ΔG°), standard enthalpy change (ΔH°), and standard entropy change (ΔS°) were calculated using Equations (10) and (11) to investigate thermodynamic modeling parameters.
(10)ΔG°=−RT lnKc,
(11)$$lnKc=\frac{\Delta H^{0}}{RT}+\frac{\Delta S^{0}}{R},$$
where *K_c_* = *q_e_*/*C_e_* is the thermodynamic equilibrium constant [58], T is the temperature (K) and R is the ideal gas constant (8.314 J/mol-K). The ΔH° and ΔS° values were calculated from the intercept and slope of Van’t Hoff’s plot (ln*K_c_* versus 1/T) illustrated in Figure 4c and are depicted in Table 4, together with ΔG° values at the considered temperatures.

The negative ΔG° values at varied temperatures indicate the spontaneous nature of the adsorption process (Table 4). In addition, the ΔG° values decreased as the temperature increased from 298 to 333 K. This suggests that, within this temperature range, a larger temperature was more favorable for MO adsorption on the CHt@MgO biocomposite. In addition, the value of ΔG° is usually employed to make a distinction between physisorption (−40 to 0 kJ/mol) or chemisorption (−400 to −50 kJ/mol). In this work, the ΔG° values were between −6.156 and −10.292 kJ/mol, which confirms that the MO adsorption onto CHt@MgO adsorbent took place through a physisorption process. The positive ΔH° and ΔS° values indicate that the MO adsorption onto CHt@MgO biocomposite was endothermic and increased the degrees of freedom of the adsorbed MO dye. In addition, the ΔH° value of 48.42 kJ/mol, which is smaller than 80 kJ/mol, confirmed that physical forces were involved in MO adsorption on CHt@MgO [59,60].

### 3.4. Comparative Performance of CHt@MgO with Other Adsorbents

For comparative study, the maximum monolayer adsorption capacity (*q_m_*) and experimental parameters of some other adsorbents reported in the literature for MO adsorption are summarized in Table 2. The results clearly display that CHt@MgO had a higher adsorption capacity (251.62 mg/g) and higher ability to absorb MO from aqueous solutions. This may be is due to the diversity in the other adsorbent’s structures and morphologies. However, the suitability as a potential adsorbent is assessed in terms of efficiency, availability, cost-effectiveness, and reusability. Hence, this study proposes CHt@MgO composite as a promising adsorbent with excellent adsorption capacity, and reusability for MO dye removal from an aqueous environment.

### 3.5. Regeneration of CHt@MgO Biocomposite

The reusability of an adsorbent is very important for the economic feasibility of the adsorption process. Therefore, in this study, the reusability of the CHt@MgO was investigated by carrying out seven adsorption/desorption cycles, as displayed in Figure 4d. It was observed that the percentage adsorption of MO dye for the first two cycles was 95.4%. Just a slight reduction in MO removal percentage was observed after the third and fourth regeneration cycles. After the fifth cycle, the percentage adsorption dropped to 90.7%, further dropping to 54.5% at the seventh regeneration cycle. Thus, it could be concluded that CHt@MgO was highly effective for MO adsorption for five consecutive regeneration cycles.

### 3.6. MO Adsorption Mechanism

According to the pH study results, the CHt@MgO was protonated and generated a positive charge on its surface at pH 6, and the MO molecule carried both positive (^+^NH(CH_3_)_2_ or ^+^N(CH_3_)_2_ or –N=N^+^H–) and negative (–SO_3_^−^) charges, depicted in Figure 1. The negatively charged –SO_3_^−^ on the MO surface was attracted to the positively charged ^+^OH_2_ on the CHt@MgO surface through an electrostatic interaction mechanism [3,4]. This led to an increase in MO adsorption capacity of CHt@MgO. According to the kinetic and isotherm modeling results, the MO adsorption mechanism onto CHt@MgO, which is schematized in Figure 5, can be summarized as follows: (i) electrostatic attraction interaction between negative charged –SO_3_^−^ present on the MO surface and positive charges of –^+^OH_2_ on the CHt@MgO surface; (ii) hydrogen bonding between –OH groups on CHt@MgO surface and nitrogen atoms of N(CH_3_)_2_ or –N=NH– on the MO surface [5]. A similar trend was observed for the adsorption of MO dye onto Fe-loaded chitosan film [61], biopolymer chitosan CHI [62], and karaya gum/chitosan sponge [63]. Abdul Mubarak et al. [61] reported that the binding mechanism of MO adsorption onto Fe–CS was via two mechanisms, namely, electrostatic attractions (between the negatively charged (SO_3_^−^) of MO molecule dye with the positively charged (NH_3_^+^) groups onto the surface of the Fe–CS and H bonding (between oxygen and nitrogen atoms of MO dye and free H onto the surface of Fe–CS). In addition, the thermodynamic parameters revealed that the MO adsorption onto CHt@MgO adsorbent took place through a physisorption process through electrostatic interaction mechanism.

## 4. Conclusions

In conclusion, a chitin-based biocomposite was successfully synthesized by incorporating MgO, characterized, and tested as an adsorbent for the removal of anionic MO dye from aqueous solutions. FT-IR and XRD analysis data confirmed the successful synthesis of CHt@MgO biocomposite. The adsorption equilibrium and kinetic results respectively fitted the Langmuir isotherm and the PFO kinetic models. The Langmuir *q_m_* value for MO adsorption on CHt@MgO was found to be 252 mg/g at 60 °C. Furthermore, thermodynamic parameters indicated that MO adsorption over CHt@MgO biocomposite was spontaneous and endothermic. Reutilization of CHt@MgO was proven to be feasible since adsorption percentages of 94% remained after four consecutive regeneration cycles. Overall, this study showed that CHt@MgO biocomposite is a very promising cost-efficient and ecofriendly material with high MO removal efficiency and excellent regeneration ability.

## Data Availability

The data are available on request.

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
