# Peer review of "Chitin-Based Magnesium Oxide Biocomposite for the Removal of Methyl Orange from Water"

_ijerph, 2023, doi:10.3390/ijerph20010831_

Round 1

Reviewer 1 Report

Hicham Majdoubi describes the properties of methyl orange adsorption by a Chitin-MgO complex. The author provided a complete characterization of the biocomposite and of its adsorption properties.

To improve the impact of the manuscript, I suggest to address the following aspects.

-          What is the amount of shrimp shell used for the extraction of chitin? Please provide this information in the methods.

-          Figure 1 fix caption.

-          Since EDX analysis is related to SEM, I suggest to include it in Figure 1.

-          Figure 3: please provide figures with appropriate resolution (Insert a and insert c).

-          3.5. MO adsorption mechanism: the author proposes an adsorption mechanism which is well described and commented. Could the author provide some references of the adsorption mechanism when possible?

Author Response

Comments file attached

Reviewer 2 Report

This paper developed a chitin-based magnesium oxide biocomposite for the removal of methyl orange from water. The adsorption parameters and characteristics were studied. Results indicate this new absorbent is economical and efficient.

1.Check the title of Fig.3.

2.Check the title of Fig.4 and the figure is not clear..

3.Please introduce the regeneration process. Regeneration should not only be analyzed with cycles in Fig. 4.

4.The adsorption mechanism should be analyzed in more detail and proofs.

5.The results should be discussed and compared with other studies.

6.How about the merits and performance of CHt@MgO compared with other researches and absorbents.

7.Please add error bars in the figures.

Author Response

Response file attached

Round 2

Reviewer 2 Report

It seemed this paper has been revised according the review points. It can be accepted for publication.